# *Rhododendron chrysanthum*’s Primary Metabolites Are Converted to Phenolics More Quickly When Exposed to UV-B Radiation

**DOI:** 10.3390/biom13121700

**Published:** 2023-11-24

**Authors:** Fushuai Gong, Wang Yu, Qingpan Zeng, Jiawei Dong, Kun Cao, Hongwei Xu, Xiaofu Zhou

**Affiliations:** Jilin Provincial Key Laboratory of Plant Resource Science and Green Production, Jilin Normal University, Siping 136000, China

**Keywords:** UV-B radiation, *Rhododendron chrysanthum*, primary metabolite, phenolic compounds, ultra-performance liquid chromatography tandem mass spectrometry (UPLC-MS/MS), ABA

## Abstract

The plant defense system is immediately triggered by UV-B irradiation, particularly the production of metabolites and enzymes involved in the UV-B response. Although substantial research on UV-B-related molecular responses in *Arabidopsis* has been conducted, comparatively few studies have examined the precise consequences of direct UV-B treatment on *R. chrysanthum*. The ultra-high-performance liquid chromatography tandem mass spectrometry (UPLC-MS/MS) methodology and TMT quantitative proteomics are used in this study to describe the metabolic response of *R. chrysanthum* to UV-B radiation and annotate the response mechanism of the primary metabolism and phenolic metabolism of *R. chrysanthum*. The outcomes demonstrated that following UV-B radiation, the primary metabolites (L-phenylalanine and D-lactose*) underwent considerable changes to varying degrees. This gives a solid theoretical foundation for investigating the use of precursor substances, such as phenylalanine, to aid plants in overcoming abiotic stressors. The external application of ABA produced a considerable increase in the phenolic content and improved the plants’ resistance to UV-B damage. Our hypothesis is that externally applied ABA may work in concert with UV-B to facilitate the transformation of primary metabolites into phenolic compounds. This hypothesis offers a framework for investigating how ABA can increase a plant’s phenolic content in order to help the plant withstand abiotic stressors. Overall, this study revealed alterations and mechanisms of primary and secondary metabolic strategies in response to UV-B radiation.

## 1. Introduction

China’s Changbai Mountain are home to the medicinal properties ornamental shrub *R. chrysanthum*. The predominant abiotic stressor in the Changbai Mountains is UV-B radiation because of the area’s unique alpine habitat, which includes cold temperatures, a thin atmosphere, and intense sun radiation. As a result, *R. chrysanthum* is a unique plant resource for the investigation of plant adaptability, and our earlier research has enough physiological support for the plant’s great tolerance to UV-B exposure.

UV-B radiation (280–315 nm) was a natural component of sunlight before the 1980s and 1990s, when a surge in the atmosphere’s hydrochlorofluorocarbon caused a severe thinning of the stratospheric ozone layer and a corresponding rapid rise in the amount of UV-B light reaching the Earth’s surface [1,2]. Obviously, this has a detrimental effect on farming and the vegetation in the natural ecosystem [3]. Even while UV-B only makes up a little portion of the energy, it has a lot of energy on its own [4]. Induced by the UV resistance motif 8 (UVR8)-dependent UV-B signaling pathway, UV-B-responsive genes encourage the formation of phenolic substances such as hydroxycinnamic acid, Kaempferol-3-hexoside rhamnose, and luteolin (5,7,3′,4′-Tetrahydroxyflavone) [5,6,7]. Numerous studies have looked at the long-term impact of UV-B radiation on plants, but few have concentrated on the short-term consequences. Studies on the impact of short-term UV-B exposure may aid in the understanding of the UV-induced pathways of signaling as well as the modifications in compounds and enzymes that occur immediately in reaction to UV irradiation. In light of this hypothesis, it is critical to comprehend how *R. chrysanthum* metabolites adjust to increased UV-B exposure.

Phenolic compounds make up the biggest class of secondary metabolites found in plants. In addition to aiding in the creation of cell walls, these substances shield plants against diseases and UV radiation [8]. Phenolics, particularly flavonoids, are the secondary metabolites that are most important for UV protection out of the more than 8000 that exist. According to Marco Santin et al., UV-B radiation triggers the downstream activation of several phenylpropane biosynthetic and regulatory genes, which causes the levels of particular phenolic compounds to significantly increase [9]. Depending on their chemical makeup, phenolic compounds can be categorized into three classes and show a variety of structural variations in nature. Specifically, there are three types of phenolic compounds: (1) those with a C6C1 carbon dioxide in the framework such as 4-hydroxybenzoic acid and 2,5-Dihydroxybenzoic acid; (2) those with C6C3 carbon bones, such as caffeic acid, p-coumaric acid, and ferulic acid; and (3) those with a C6C3C6 carbon skeleton (flavonoids are the typical C6C3C6) [10,11]. Higher plants produce secondary metabolites through the shikimate pathway, which is also the pathway by which phenolic compounds are often generated. Phosphoenolpyruvate and erythrose 4-phosphate are the starting points of the route, whereas chorismite is its last stop [12,13]. A vital metabolite produced by the chorion is phenylalanine [14]. In plant metabolism, phenylalanine is regarded as a universal precursor of the molecules C6C1-, C6C3-, and C6C3C6 and their polymers. The phenolic compound buildup, which is controlled by abiotic stressors, has received a lot of attention in studies. A possible explanation for the large carbon flux through the phenylalanine metabolic network is the vast range of physiological functions of these substances and their significance for plant growth, development, defense, and environmental reactions [15]. There are many phenolic chemicals in *R. chrysanthum* that can be studied.

Abscisic acid (ABA), a plant hormone, is crucial for numerous physiological processes, including seed germination, dormancy, growth, and adaptation to biotic and abiotic stressors. By controlling the processes of starch breakdown, source pool transfer, and leaf senescence, ABA may encourage bud dormancy. Additionally, ABA shields plants from dry stress by encouraging the build-up of proline. It was demonstrated that *TaFDL2-1A* enhances transgenic wheat’s resistance to drought by positively regulating ABA production, ABA responsiveness, and ROS scavenging [16].

Metabolomics is a rapidly developing field that plays a critical role in understanding the complicated primary and secondary metabolite variations in higher plants [17]. Modern biology is increasingly adopting and recognizing metabolomics techniques. Nuclear magnetic resonance spectroscopy (NMR), gas chromatography–mass spectrometry (GC-MS), and liquid chromatography–mass spectrometry (LC-MS) are typical analytical methods used in plant metabolomics research [18,19]. Due to recent advancements in UPLC-MS techniques, which offer greater and fewer run durations than traditional HPLC-MS techniques, metabolite analysis has become more reliable [20]. UV-B stabilizes health-promoting bioactive chemicals that are necessary for the bread-making process [21]. Additionally, the analysis of flavonoids using ultra-high-performance liquid chromatography revealed that UVR8-mediated flavonoid induction is a UV-B tolerance feature that is conserved in terrestrial plants and could be an early adaptation to life on Earth [22].

UV-B light activates a number of phenolic compound biosynthesis enzymes, including chalcone synthase (CHS, E.C. 2.3.1.74) and phenylalanine deaminase (PAL, E.C. 4.3.1.25) [23]. The enzymes PAL, cinnamate 4-hydroxylase (C4H, E.C. 1.14.13.11), and 4-coumarate coenzyme A ligase (4CL, E.C. 6.2.1.12) are engaged in the upstream phenylalanine route, which provides precursors for the downstream flavonoid metabolism pathway as well as the flavonol metabolism pathway.

In general, plants’ primary defense against harmful UV-B radiation is to encourage the creation of UV-absorbing substances such phenolic compounds, flavonoids, and hydroxycinnamates [24]. The mechanisms of *R. chrysanthum*’s metabolic reactions to UV-B light are poorly understood. We investigated the impact of short-duration UV-B rays on the formation of both primary and phenolic compounds specific for primary and phenolic metabolites in *R. chrysanthum* using ultra-performance liquid chromatography–mass spectrometry (UPLC-MS) metabolomics in order to investigate in depth the reactions and mechanisms of how metabolic reprogramming reaches a new homeostasis after UV-B light.

## 2. Materials and Methods

### 2.1. Plant Material, Growing Conditions, and Treatments

Three of the nine (n = 3) genetically identical *R. chrysanthum* seedlings that were chosen were separated into three groups as an array of biological duplicates. Before receiving UV-B therapy, all seedlings were cultivated in a sophisticated artificial climate room (http://www.nbsaifu.cn/about_about/gsjj9f6.html accessed on 15 July 2023; Ningbo Saifu Experimental Instrument Co., Ningbo, China) at a temperature of 25 °C/25 °C (day/night) and a photoperiod of 14 h/10 h (day/night). One batch of seedlings was grown in a media that also contained exogenous ABA, whereas the remainder of the two groups were both cultured in regular *R. chrysanthum* medium. After four months of growth, all three groups of seedlings were at once moved to an artificial climate chamber used for this experiment that replicated an alpine environment (18 °C (light for 14 h)/16 °C (dark for 10 h)) with white light radiation of 50 mol (photons) m^−2^s^−1^ and 60% relative humidity. Group M of seedlings grown on regular *R. chrysanthum* media were the only ones to receive PAR light. Group N of seedlings grown on regular *R. chrysanthum* medium were only exposed to UV-B light [25]. For group Q, seedlings grown in media with exogenous ABA application likewise only received UV-B light. The three groups’ seedlings underwent two days of artificial climate chamber incubation with 8 h of light per day.

In a simulated climate room, UV-B light (UV-B) (295–315 nm) and PAR (400–700 nm) were applied to three sets of seedlings [26]. UV intensity measurements were made by Sentry Optronics Corp. in China. The culture flasks were specifically covered during the UV-B treatment using a 295 nm long-pass filter (Filter Long 2IN SQ Filter, Edmund, NJ, USA). Above the culture flasks, 400 nm long-pass filters were installed for the PAR treatments (Edmund). According to the transmission function of the long-pass filter, the samples actually received an irradiance of 50 mol (photons) m^−2^ s^−1^ of PAR and 2.3 W m^−2^ UV-B, respectively.

### 2.2. Identification and Quantification of Primary Metabolites and Secondary Metabolites by UPLC-MS/MS

The following steps were followed when conducting the metabolomic analysis with Metware Biotech Inc. in Wuhan, China: At −20 °C, 1200 μL of an aqueous solution of 70% methanol was used to extract 50 mg of crushed, lyophilized powder. The supernatant was aspirated and put through a 0.22 μm membrane after being centrifuged at 12,000 rpm for 3 min. The metabolites in samples of *R. chrysanthum* were broadly quantified using a liquid chromatography–electrospray ionization tandem mass spectrometry (LC-ESI-MS/MS) technology. Following the established techniques previously described, Wuhan METWARE Biotechnology Co., Ltd. in Wuhan (Wuhan, China) employed an LC-ESI-MS/MS system (UPLC, gasket-packed UFLC CBM30A, Shimadzu, Kyoto, Japan; MS, 6500 QTRAP, Applied Biosystems, Norwalk, CT, USA), and analyses were performed on all sample extracts [27,28]. Ultrapure water with a formic acid content of 0.1% and acetonitrile with a formic acid content of 0.1% were the two mobile phases A and B, respectively. According to the gradient program, phase B was raised from 5% to 95% in 9 min, held for 1 min, and then immediately lowered to 5% and adjusted for 14 min. The column temperature was 40 °C, the mobile phase flow rate was 0.35 mL/min, and the injection volume was 2 μL.

By using electrospray ionization (ESI), metabolites were ionized. They were recognized using the Metware database’s secondary fragment ion spectra. By using triple quadrupole mass spectrometry, the identified metabolites were measured through multiple reaction monitoring. The relative content of metabolites was calculated using peak regions. Variable importance (VIP) in projection scores with orthogonal partial least squares discriminant analysis (OPLS-DA) > 1 and |fold change| > 1.2 were regarded as differential metabolites (DMs). Based on the KEGG database, metabolites were enhanced and categorized.

### 2.3. Analysis of Metabolite Data

The statistics function prcomp in R (www.r-project.org, accessed on 15 July 2023) was used to perform unsupervised PCA (principal component analysis). Prior to unsupervised PCA, the data were unit-variance-scaled. Using the program MetaboAnalyst 5.0 (https://www.metaboanalyst.ca/, accessed on 17 July 2023), metabolite correlation network diagrams were created for the metabolites in the samples. Utilizing the KEGG Compound database (http://www.kegg.jp/kegg/compound/, accessed on 20 July 2023), identified metabolites were annotated. The annotated metabolites were then linked to the KEGG Pathway database (http://www.kegg.jp/kegg/pathway.html, accessed on 25 July 2023). The *p*-values for the hypergeometric test were used to establish the importance of the pathways where significantly regulated metabolites had been mapped.

### 2.4. Identification and Quantification of Proteins by Liquid Chromatography–Mass Spectrometry Analysis

In order to execute the proteome analysis, the following procedures were followed: protein extraction, trypsin digestion, Tandem Mass Tag (TMT) tagging, peptide fractionation, LC-MS/MS analysis, protein annotation, and functional enrichment [29].

### 2.5. Statistical Analysis

R (http://cran.r-project.org/, accessed on 2 July 2023) and IBM SPSS statistical software (https://www.ibm.com/analytics/spss-statistics, IBM SPSS Statistics 26, accessed on 5 July 2023) were used for the statistical analysis. Results show letters to indicate the levels of significance. A one-way analysis of variance (ANOVA) and a Pearson correlation test with an 80% threshold of significance were used to evaluate the data.

## 3. Results

### 3.1. The Primary Metabolism Reprogramming in the Presence of UV-B Stress

The repeatability of various experimental materials under UV-B exposure was ensured by inserting quality control (QC) samples into every 10 test samples during the identification of metabolites using the UPLC-MS/MS method [30]. The curves of the total ion current for metabolite detection showed high overlap when the total ion current plots (TIC plots) of various QC samples were analyzed by mass spectrometry detection (Appendix A). This means that the retention times and peak intensities were consistent, indicating that mass spectrometry had better signal stability when detecting the same sample at various times.

To comprehend the general metabolic variations and variability between groups among the samples, Principal Component Analysis (PCA) was carried out on the specimens, including the QC samples. PCA determines whether there are variations among metabolomes by separating metabolome trends into groups. Insignificant differences were found between UV-B pressure treatment specimens of the same materials in PCA of the control, experimental, and QC samples; however, there was a clear pattern of segregation between the various groups (Figure 1a). The Pearson correlation coefficient (PCC) R was employed as an evaluation metric for the correlation of the three biological replicates. Heat maps of the correlation between the three biological replicates and the QC samples were also created. The data showed a strong positive association among the three biological replicates (Figure 1b).

The MetWare Database (MWDB), multiple reaction monitoring (MRM), and the Kyoto Encyclopedia of Genes and Genomes (KEGG) Compound Database were used in the qualitative and quantitative mass spectrometry analysis of the metabolites in each UV-B stress sample. Comprehensive targeted metabolic analysis yielded a total of 2148 metabolites, of which 615 were primary metabolites, comprising organic acids, lipids, amino acids and their derivatives, nucleotides and their derivatives, and other classes (Figure 1c). The majority of amino acids and their derivatives, such as L-phenylalanine, L-tyrosine*, etc., were upregulated in the MN group as well as in the NQ group. Of these, phenylalanine attenuates the adverse effects of abiotic stresses. While nucleotides and their derivatives were upregulated in the MN group but did not show any significant changes in the NQ group, the majority of lipid metabolites did not exhibit any appreciable changes. The same trend was also observed in the organic acids, such as 3-Dehydroshikimic acid and L-Tartaric acid. The 3-Dehydroshikimic acid substantially altered in response to UV-B exposure, indicating that it functions as a crucial intermediary product influencing the metabolic pathways involved in the biosynthesis of aromatic amino acids. The majority of the saccharides, including D-Lactose* and DMelezitose O-rhamnoside, among other classes, were downregulated in the MN group but not significantly altered or upregulated in the NQ group; 1359 different secondary metabolites were found, such as phenolic acids, flavonoids, quinones, lignans and coumarins, ellagitannins, alkaloids, terpenoids, steroids, and others (Figure 1d). Significant changes were observed in the phenolic acids and flavonoids. For example, phenolic acid metabolites such as 4-O-Galloylquinic acid, chlorogenic acid (3-O-Caffeoylquinic acid), methyl caffeate, etc. were upregulated in the MN group and also tended to be upregulated in the NQ group.

A supervised OPLS-DA model was also used to compare metabolite levels and pinpoint the factors that contributed to group disparities. The OPLS-DA model was used to compute the differences between M and N (R^2^X = 0.732, R^2^Y = 1, Q^2^ = 0.7) (Figure 1e) and between N and Q (R^2^X = 0.442, R^2^Y = 0.998, Q^2^ = 0.643) (Figure 1f). Both comparisons had Q^2^ values that were more than 0.5, showing that the two models were steady and had distinct metabolic phenotypes in response to UV-B exposure. These findings imply that the metabolite composition of *R. chrysanthum* is significantly altered by UV-B stress.

A total of 56 significantly modified metabolites were found between N and Q (39 upregulated, 17 downregulated) (Figure 2a,c), and 203 were found between M and N (126 upregulated, 77 downregulated) (Figure 2a,b) using the thresholds of |log2Fold Change| ≥ 1.2 and VIP (variable importance in project, VIP) > 1. Under UV-B stress, groups M and N as well as groups N and Q only co-influenced eighteen DMs, with seventeen of these co-influencing compounds located in the upper Venn diagram (Figure 2e), including S-methyl-L-cysteine and 6-O-methylguanine, and just one in the lower Venn diagram (Figure 2f), namely, Nystose. This is due to the fact that the differentially expressed metabolites that are most frequently impacted in both Venn diagrams are those metabolites whose expression levels varied between the two groups (Figure 2d). Our study focused on the DMs of these two trends, where the first trend’s DMs were those that responded favorably to UV-B stress and the second trend’s DMs were those that responded adversely to UV-B stress. The differential compounds of the two trends were classified as core DMs. Amino acids and their derivatives, which are the main primary metabolites of plants and, as monomers, are crucial to the synthesis of proteins, are the most varied group of core differential metabolites. The categorization and expression data for the 18 core DMs are shown in Figure 2g, together with information about the core DMs (Appendix A), Table 1 shows information on the first seven core DMs.

### 3.2. Dynamics of Phenolic Compounds under UV-B Radiation

The screening parameters for secondary metabolites matched those for primary metabolites. The MN group of secondary metabolites had a total of 392 DMs that were upregulated and 88 which were downregulated, while the NQ group had 180 DMs that were upregulated and 136 that were downregulated (Figure 3a). According to the VIP value, Figure 3b,c show the first 15 metabolites were up- and downregulated in the MN group and the NQ group, respectively. In addition, following UV-B stress and ABA addition, cluster analysis revealed three distinct metabolite trends (Figure 3d). Naturally, we focused more on the sixth cluster. There, we discovered that the DMs that were clustered to the sixth cluster showed a growing trend in the MN group, indicating that these metabolites are crucial in the response to UV-B stress, and continued to show an upward trend in the NQ group, showing that the outwardly applied ABA helps the *R. chrysanthum* to withstand UV-B stress. The majority of the secondary DMs in cluster VI are phenolic substances. As a result, we suggest that variations in phenolic content may help to improve the fluidity of membranes and promote the protective effects of reactive oxygen species (ROS) when exposed to UV-B stress. There were statistical differences between groups in the quantity of phenolic chemicals (Figure 3e). It was discovered that *R. chrysanthum* was able to withstand UV-B stress thanks to the external application of ABA, which also caused phenolic abundance to grow greatly following UV-B stress.

The ratio of the phenolic content in *R. chrysanthum* after UV-B radiation only to the phenolic content in *R. chrysanthum* without UV-B radiation as well as the ratio of the phenolic content in *R. chrysanthum* after UV-B radiation with externally applied ABA to the phenolic content in *R. chrysanthum* after UV-B radiation with no ABA applied were greater than one (Figure 4a–c). In other words, phenolics contribute to *R. chrysanthum*’s ability to fend off UV-B harm. The plant’s ability to withstand UV-B stress was also improved by the effective promotion of stomatal closure by ABA when administered externally. Additionally, the abundance changes of all C6C3C6 backbone phenolics were less pronounced in the MN group than in the NQ group, indicating that this group of phenolics likely contributes significantly to *R. chrysanthum*’s UV-B protection (Figure 4c).

### 3.3. Network Diagram and Correlation Analysis of R. chrysanthum’s Main and Phenolic Metabolite Relationships

A metabolite interaction network was carried out to further investigate how the primary and phenolic metabolites in the *R. chrysanthum* responded to UV-B exposure. To supervise the network’s simplification, the Degree Filter was set to 2.0 and the Betweenness Filter was set to 1.0. Adenine is a significant “hub” in the network since it has the greatest degree and the most connections to other nodes (84). The purine metabolism pathway is the one that is reportedly damaged, according to the network diagram. With a degree of 71, L-phenylalanine is also a more significant “hub” in the network, influencing pathways such as the biosynthesis of phenylalanine, tyrosine, and tryptophan as well as the metabolic pathway of phenylalanine. Both of these pathways have *p*-values of 0.01, and L-phenylalanine is labeled in the figure with its structural formula. L-tyrosine’s degree is also relatively large, 63, and plays an important role in the network, affecting pathways. In addition to the biosynthesis of phenylalanine, tyrosine, and tryptophan, there is also the tyrosine metabolic pathway, which is labeled as the L-tyrosine structural formula in the figure. Many plant secondary metabolites, such as rhodiolosides, betalains, and tocopherols (vitamin E), are derived from tyrosine.

For each category of phenolic core DMs and each category of primary core DMs in M, N, and Q, Pearson’s pairwise correlation coefficient values were obtained. Significant correlations were defined as those with a *p*-value of 0.05 or below and a relationship analysis threshold of 0.8.

In the correlation analysis between amino acids and their derivatives and phenolic acids, there were 57 positive correlations and no negative correlations (Figure 5b). For instance, L-phenylalanine showed a positive correlation with ferulic acid-4-O-glucoside, 3-O-Feruloylquinic acid-O-glucoside, and particularly with Trans-5-O-(p-Coumaroyl) shikimate, as well as significantly positive correlations with chlorogenic acid (3-O-Caffeoylquinic acid) and correlation coefficients of 0.95 and 0.90, respectively. It was demonstrated (Figure 5c) that there were 40 positive correlations and, once more, no negative correlations in the correlation study of amino acids and their derivatives with flavonoids. For instance, L-tyrosine* showed a strong association with 5,7,3′-Trihydroxy-4,6,5′-trimethoxyxanthone, luteolin-7-O-glucuronid e-5-O-rhamnoside, and especially with dihydromyricetin (ampelopsin), which had a correlation value of 0.86. There were 10 positive associations and no negative correlations in the correlation study of amino acids and their derivatives with lignans, alkaloids, and terpenoids (Figure 5d). For instance, N-(2-hydroxy-4-methoxyphenyl) acetamide was strongly positively associated with L-hom ocysteine, Dimethylfraxetin, Dehydrodiconiferyl alcohol, and others, with a correlation coefficient of 0.96. Combining these three correlation analyses, it was discovered that amino acids and their derivatives significantly correlated with phenolic acid compounds, and the interaction network was more complex. This finding suggests that amino acids and their derivatives, along with phenolic acids, play significant roles in the *R. chrysanthum*’s defense against UV-B stress and that the two classes of compounds can be used as markers for assessing the *R. chrysanthum*.

Again, only 25 positive correlations and no negative correlations were found in the correlation analysis of nucleotides and their derivatives with phenolic acids (Figure 5e). For instance, with correlation values of 0.92 and 0.88, respectively, adenine strongly positively linked with 2-phenylethanol, and xanthosine significantly positively associated with Trans-5-O-(p-Coumaroyl) shikimate. Only 22 positive correlations between nucleotides and their derivatives and flavonoids were found in the correlation study (Figure 5f). For instance, adenine and dihydromyricetin (ampelopsin) had a correlation of 0.91, which was significantly positive. In addition, 6-O-methylguanine and dihydromyricetin (ampelopsin) exhibited positive correlations with adenine and 13(S)-HODE, respectively. There were only seven positive connections and no negative correlations in the correlation study of nucleotides and their derivatives with lignans, alkaloids, and terpenoids (Figure 5g). For instance, adenine and N-(2-hydroxy-4-methoxyphenyl) acetamide had a correlation coefficient of 0.94 that was significantly positive. Combining the results of the three correlation analyses, it was discovered that nucleotides and their derivatives had the strongest link with phenolic acids and interacted with the most compounds, followed by flavonoids.

Thirty-three correlations were found in the correlation analysis of vitamins, lipids, sugars, and phenolic acids (Figure 5h), twenty-nine of which were positive and four of which were negative. For instance, Nystose and methyl caffeate were negatively connected, while 13(S)-HODE and Elaidolinolenic Acid were positively correlated. Additionally, there was a substantial positive link between chlorogenic acid (3-O-caffeoylquinic acid) and nicotinic acid (Vitamin B3), with correlation coefficients of 0.88 and 0.90, respectively. Twenty-six correlations were found in the correlation analysis of vitamins, lipids, sugars, and flavonoids (Figure 5i), of which twenty-two were positive and four were negative. Specifically, dihydromyricetin (ampelopsin) and nicotinic acid (Vitamin B3) revealed a substantial positive link with a correlation of 0.94. Nicotinic acid (Vitamin B3) was positively correlated with 13(S)-HODE, Nystose was negatively correlated with luteolin-7-O-glucuronide-5-O-rhamnoside, etc. Only seven associations, four of which were positive and three of which were negative, were discovered when vitamins, lipids, and carbohydrates were correlated with lignans, alkaloids, and terpenoids (Figure 5j). It was found that 13(S)-HODE and N-(2-hydroxy-4-methoxyphenyl) acetamide had positive and negative correlations, respectively. With a correlation coefficient of 0.95, N-(2-hydroxy-4-methoxyphenyl) acetamide in particular showed a highly positive link with nicotinic acid (Vitamin B3). These three correlation analyses combined showed that there were more metabolite interactions between vitamins, lipids, and sugars in correlation with phenolic acids to help *R. chrysanthum* resist UV-B stress, indicating that phenolic acid compounds can be used as markers for *R. chrysanthum*’s reaction to UV-B stress.

When viewed collectively, the nine correlation analyses demonstrate the critical role that amino acids and their derivatives play in the network of correlation analyses as well as the complex interactions that phenolic compounds, including phenolic acids and flavonoids, exhibit with other compounds in this network. Additionally, it was shown that only sugars, lipids, and vitamins had negative correlations with other substances, which may be connected to Nystose’s decreased expression in response to UV-B radiation. Together, our findings imply that in response to extreme UV-B exposure, *R. chrysanthum* mobilizes more complicated metabolite relationships to cooperate.

### 3.4. Building a Comprehensive Picture of the Primary and Phenolic Metabolite Networks of the R. chrysanthum

Simplified primary and phenolic compound metabolic pathways were employed to demonstrate the *R. chrysanthum*’s metabolic response to UV-B radiation in order to more clearly illustrate the linkages between the data mentioned above. Heat maps are used to display the expression levels of the relevant enzymes in the proteome in an effort to completely reveal the enzymes associated with the L-phenylalanine and flavonoid biosynthesis pathways as well as in the monolignol biosynthesis pathway (Figure 6a). Metabolite abundance and the levels of metabolite-related enzyme expression in metabolic pathways are intricately connected. Some amino acids, such as S-methyl-L-cysteine, L-phenylalanine, and L-tyrosine*, considerably elevated in response to UV-B radiation for primary metabolites. Moreover, following UV-B treatment, the concentrations of acids implicated in the tricarboxylic acid (TCA) cycle, such as fumarate, malate, and succinate, were elevated. In terms of the phenolic components, Trans-5-O-(p-Coumaroyl) shikimate and naringenin chalcone were the primary phenolic acids and flavonoids that were upregulated in *R. chrysanthum*. Furthermore, the shikimate acid pathway uses L-phenylalanine as a crucial node in the synthesis of phenolic compounds, which may explain why most phenolic compounds rise in concentration when exposed to UV-B light. The boxed line graphs show how the expression of L-phenylalanine, naringenin chalcone, and Trans-5-O-(p-Coumaroyl) shikimate changes in response to UV-B stress. Significant changes in the expression of L-phenylalanine and naringenin chalcone can be observed, indicating a potential protective effect of *R. chrysanthum* against UV-B stress.

### 3.5. Enrichment Pathway Analysis and Correlation between R. chrysanthum Proteomics and Metabolomics in Response to UV-B Stress

KEGG enrichment analysis of the 11 identified enzymes was performed on a total of eight metabolic pathways (Figure 7a,b). These included flavonoid biosynthesis; circadian rhythm of plant, propane, piperidine and pyridine alkaloids biosynthesis; secondary metabolites biosynthesis; metabolic pathways, flavonoids, and flavonoids biosynthesis; phenyl metabolic pathway, piperidine, and pyridine alkaloids biosynthesis; secondary metabolites biosynthesis; metabolic pathways, flavonoids, and flavonols biosynthesis; phenylalanine, tyrosine, and tryptophan biosynthesis; and amino acid biosynthesis. According to the results, L-phenylalanine, naringenin chalcone, and Trans-5-O-(p-Coumaroyl) shikimate increased in response to UV-B stress while cinnamic acid, p-Coumaric acid, and naringenin (5,7,4′-Trihydroxyflavanone) * were continually depleted. Among these, Trans-5-O-(p-Coumaroyl) shikimate, naringenin chalcone, and L-phenylalanine displayed considerably higher levels and are important nodes in the UV-B stress regulation network.

Pearson correlation studies were carried out (Figure 7c) to further evaluate the links between important enzymes and essential DMs. The hub of the entire network, CHS and CHI, exhibits a favorable connection with metabolites. The 2-hydroxy-3-phenylpropanoic acid, isorhamnetin-3-O-gallate, and 3-O-Feruloylquinic acid-O-glucosid all showed strong positive associations with CHS1 and CHS, with the least significant correlation coefficient being 0.88 and the most significant correlation coefficient being as high as 0.99. With correlation coefficients around 0.9, CHI and CHI3 demonstrated valuable positive relationships with 3-O-Feruloylquinic acid-O-glucosid, 2-hydroxy-3-phenylpropanoic acid, and L-phenylalanine. Two metabolites, 1-O-p-Coumaroyl-D-glucose and 3-hydroxy-3-methylpentane-1,5-dioic acid, are under the control of F3H. Additionally, PPA-AT favorably regulated 1-O-p-Coumaroyl-D-glucose and 2-hydroxy-3-phenylpropanoic acid. These six metabolites belonged to amino acids and their derivatives, phenolic acids, and flavonoids, indicating that CHS, CHI, F3H, and PPA-AT collectively affected primary metabolites as well as phenolic compounds against UV-B stress in the *R. chrysanthum*. This further affected phenylpropane biosynthesis and controlled the primary as well as phenolic compound metabolic pathways. The information about 11 important enzymes is shown in Table 2.

## 4. Discussion

Numerous responses of plants to elevated UV-B radiation have an impact on plant development, growth, and the accumulation of secondary metabolites [31]. According to the UPLC-MS/MS detection platform, the abundance and characterization of metabolites were investigated in this study, and *R. chrysanthum*’s response to UV-B stress was examined at the level of both primary and secondary metabolites. These analyses together revealed alterations to the metabolic profiles of various metabolites of the *R. chrysanthum* under UV-B stress and explored the reprogramming of primary and secondary metabolic responses to UV-B. In order to validate the changes in the expression of important metabolites in the pathway in response to UV-B stress, we also examined protein expression under UV-B stress based on LC-MS/MS and evaluated changes in the abundance of the key enzymes of L-phenylalanine, tyrosine, and tryptophan; the flavonol metabolism pathway, the flavonoid metabolism pathway, and the monolignol biosynthesis pathway, as well as changes in the pathways of enzyme action through the proteome.

Based on UPLC-MS/MS metabolomics, we examined specific variations in the principal metabolites of *R. chrysanthum* under UV-B radiation. Growth requires energy from sugar, and glucose is a significant soluble sugar and signaling molecule that clearly plays a fundamental role in plant stress [32]. Following UV-B radiation, the *R. chrysanthum*’s decreased glucose levels may be linked to a more pronounced transition from carbon absorption to carbon storage [31]. Additionally, larger concentrations of monosaccharides from the pentose phosphate pathway (PPP), such as D-erythrose-4-phosphate, were found in the UV-B-treated group, which may be due to the PPP pathway’s higher respiratory frequency in the *R. chrysanthum*. Through the glycolytic pathway and the TCA cycle, carbohydrates like sucrose can be oxidized, and since glycolysis can result in an increase in ATP content, the plant’s capacity to adapt to its surroundings under abiotic stress circumstances is unavoidably improved [33]. The fundamental metabolites that plants primarily produce are amino acids, which are crucial for protecting them from abiotic stressors and, more critically, serving as precursors to secondary metabolites that increase plant resistance to UV-B radiation. This research uses L-phenylalanine as an example, which exhibits a substantial rise under UV-B stress. The flavonoid pathway is indirectly impacted by the activity of pathways involving phenylpropanoid substances linked with amino acid metabolism, as has been shown in prior investigations, and the same results have been drawn in this paper [34].

The amount of flavanols and flavonols in *R. chrysanthum* increased in response to UV-B radiation, according to the dynamic response pattern of the phenolic compounds. According to physiological research, phenolics play a role in UV-B protection [35]. Within the phenolics, flavonoids and flavonols shield the cells as they collect in the *R. chrysanthum*’s epidermis, acting as filters and absorbing the majority of UV-B radiation [36,37]. Ryan et al. reported that UV radiation caused the production of flavonols with a significant amount of hydroxylation in the *Petunia* and the *Arabidopsis thaliana*. They hypothesized that flavonols might act on an as-yet-uncharacterized role in the UV stress reaction due to their UV-absorbing qualities and that the hydroxylation of flavonols may have a positive impact on antioxidant capacity [38]. Thus, under UV-B radiation, the biggest increases in the concentrations of flavonoid and flavonol substances such as luteolin (5,7,3′,4′-Tetrahydroxyflavone) and quercetin-3-O-rhamnoside (quercitrin) were seen. Additionally, it was found that C6C3C6 carbon compounds were accumulating at a faster rate than C6C3 carbon compounds like cinnamic acid and ferulic acid, which were shown to be accumulating at a slower rate. The greater UV-B absorption and antioxidant activity of C3C6C3 carbon compounds compared to C3C6 carbon compounds (Figure 4a–c) may be the cause of this event, which led to the incorporation of C6C3C6 carbon components at the expense of C3C6 carbon compounds.

For the *R. chrysanthum*, interaction network diagrams were created for the primary and phenolic metabolites, and correlation analysis diagrams were examined between each of the primary core difference metabolites and the secondary phenolic metabolites in order to better understand the differences in the specificity of metabolites identified between the UV-B-treated and control groups (Figure 5). The network diagram was mainly enriched with phenylalanine, tyrosine, and tryptophan biosynthesis as well as phenylalanine biosynthesis pathways, suggesting the significance of L-phenylalanine in the *R. chrysanthum*’s defense against UV-B stress; and the absence of phenolics-annotated pathways may be due to the fact that some of the detected phenolics were not annotated to the KEGG data, which can be further explored in future studies. A diagram of the primary and phenolic metabolite pathways was created (Figure 6) to help further clarify the mechanisms of primary as well as phenolic metabolites against UV-B damage. UV-B light causes a rise in TCA cycle intermediates such as fumarate, a key regulator of carbon and nitrogen interactions. After plants mobilize their stored carbon reserves to boost flux along the phenylpropane pathway, it appears highly likely that fumarate will be elevated under UV-B light [39]. The same conclusion was obtained in our experiments. Shikimic acid is the initiator of secondary metabolism in higher plants, linking sugar metabolism to polyphenol metabolism, and is an essential pathway for the biosynthesis of aromatic compounds. In plants, phenylalanine is recognized as a general precursor of C6C1-, C6C3-, and C6C3C6 compounds and their polymers such as tannins and lignin [40]. Thus, taken as a whole, these data imply that in response to UV-B, there is an “initiating” mechanism at the primary metabolic level in plant cells that involves the reprogramming of metabolism to effectively transfer carbon to aromatic amino acid precursors of the phenylpropane pathway [41]. Tryptophan and phenylalanine compete for chorismate to synthesize alkaloids and phenols, respectively [31]. Phenylalanine accumulated greater under UV-B radiation, according to the findings. This demonstrates that the increase in phenolic compounds during UV-B radiation, which is caused by the movement of carbon in primary metabolism, is the result of metabolic reprogramming. Phenylalanine, a critical node in phenolic metabolism, dramatically increased under UV-B radiation.

Proteomics was used to validate the aforementioned inferences made from metabolomics (Figure 7a–c). On the one hand, qualitative and quantitative analyses of the key enzymes in the metabolic pathway showed that the changes between the key enzymes in the pathway and the key metabolites showed consistency; specifically, the key enzymes in the metabolic pathway were significantly increased in the *R. chrysanthum* against UV-B stress, which in turn affected the increase in the expression of the related metabolites and the response of the entire pathway to UV-B stress. The findings of a correlation analysis of 11 major enzymes from mid-species with the core DMs, on the other hand, showed that the proteome and metabolome responded consistently to UV-B stress.

We also discovered that after applying ABA externally and then applying UV-B radiation, alterations in significant primary metabolites and phenolics were considerably more pronounced than they were after the standard UV-B radiation treatment. In contrast to flavonoids, which are secondary metabolites with antioxidant activity and UV absorbers, ABA enhances UV-B stress acclimation by promoting ROS scavenging, according to earlier investigations [42]. The results of the present study’s results, which had a much greater flavonoid content, imply that externally administered ABA may work in concert with UV-B to encourage the conversion of primary metabolites to phenolic compounds.

## 5. Conclusions

Using ultra-performance liquid chromatography tandem mass spectrometry (UPLC-MS/MS) technology, we were able to determine the metabolic profiles and explore the particular metabolic pathways of *R. chrysanthum* in response to short-duration UV-B radiation. Our findings indicate that plant cells reprogrammed the L-phenylalanine biosynthesis pathway to undergo a more intensive shift from carbon absorption to carbon accumulation as a main metabolic level mechanism in response to UV-B exposure. This gives phenylalanine, a precursor substance, a solid theoretical foundation for research on its potential to assist plants in coping with abiotic stressors. Simultaneously, alterations in pertinent metabolites and enzymes within the flavonoid and monolignol biosynthesis pathways were initiated at the secondary metabolic level. In phenolics, C6C1 carbon compounds (phenolic acids) and C6C3C6 carbon compounds (flavonoids and flavonols) were upregulated at the expense of C6C3 carbon compounds in order to withstand UV-B stress. Furthermore, we discovered that *R. chrysanthum*’s resistance to UV-B was markedly improved and that phenolic abundance increased significantly following the external application of ABA; in other words, external application of ABA may encourage the conversion of primary metabolites to phenolics in concert with UV-B, thereby strengthening the plant’s capacity for both UV-B uptake and antioxidant defense. In addition to offering insights for future research on the mechanism by which externally applied ABA can mitigate UV-B damage in the *R. chrysanthum*, this study advances our knowledge of how primary and secondary metabolites in this plant respond to UV-B stress.

## Figures and Tables

**Figure 1 biomolecules-13-01700-f001:**
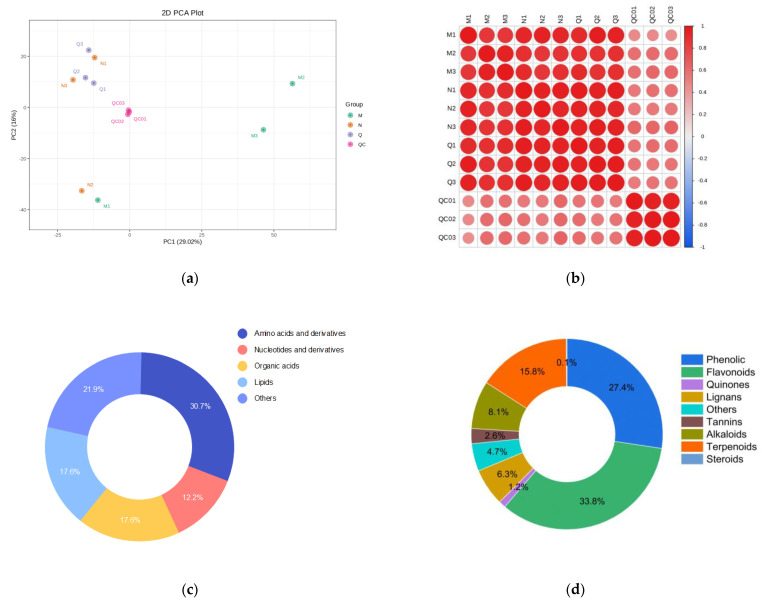
UV-B-treated as well as externally applied ABA UV-B-treated samples and quality control (QC) samples were examined in different groups and categorized for primary and secondary metabolites. (**a**) Comparing control, UV-B-treated, and quality control (QC) samples using principal component analysis (PCA); (**b**) comparison of control, UV-B-treated, and quality control (QC) data with those from various groups; (**c**) classification and proportions of primary metabolites; (**d**) classification and proportions of secondary metabolites; (**e**,**f**) the variables that led to differences between groups were monitored and calculated using the OPLS-DA model. The explanatory rates of the X and Y matrices are denoted by R^2^X and R^2^Y, respectively. Q^2^Y represents the model’s ability to forecast. The model is more stable and reliable the closer its value is near one. Additionally, Q^2^Y > 0.5 and Q^2^Y > 0.9 are regarded as valid and outstanding models, respectively.

**Figure 2 biomolecules-13-01700-f002:**
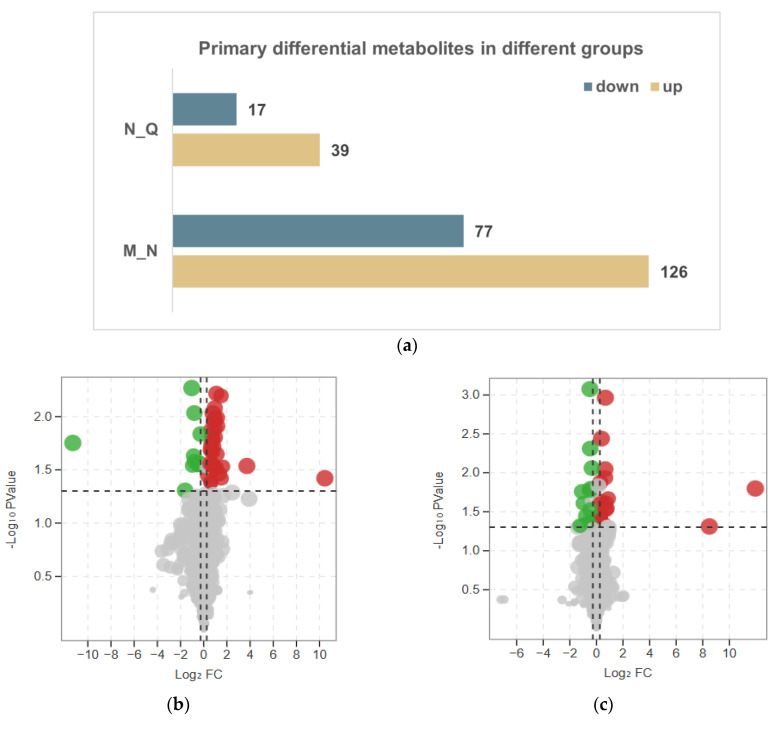
Analysis of the primary DMs in *R. chrysanthum* following UV-B exposure and UV-B exposure with externally administered ABA. (**a**) Statistic differences in metabolites across groups; (**b**,**c**) DMs of the M−N and N−Q groups are displayed on volcano plots, respectively. In the volcano plot, red circles represent up-regulated substances and green circles represent down-regulated substances, horizontal dashed lines indicate screening *p* < 0.05, vertical dashed lines screening |fold change| > 1.2; (**d**–**f**) Venn diagrams for DMs, upregulated DMs, and downregulated DMs are shown for the M−N and N−Q groups, respectively; (**g**) The three MNQ groups’ core DMs are shown in bar graphs.

**Figure 3 biomolecules-13-01700-f003:**
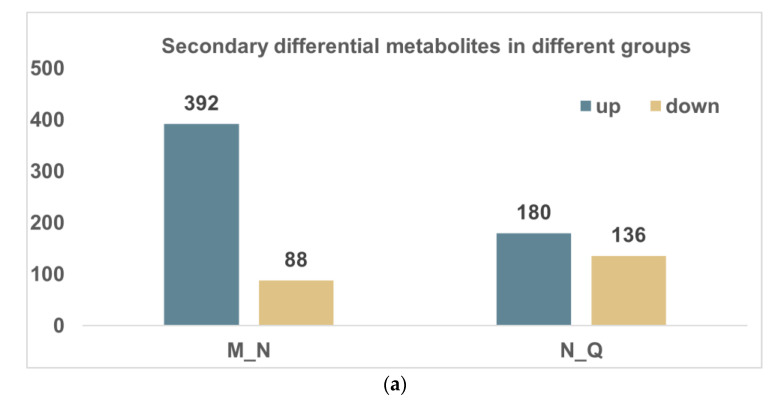
Secondary *R. chrysanthum* metabolites in reaction to UV-B light and UV-B radiation with externally administered ABA. (**a**) Comparative metabolite histogram statistics between different categories. (**b**,**c**) For the M−N and N−Q groups, respectively, the top 15 different metabolites ranked by VIP value of secondary differential compounds are displayed, with red representing high levels and green indicating low levels; (**d**) analyses of metabolites using the K-means clustering technique; (**e**) trends in phenolic metabolite abundance with UV-B therapy and exposure to UV-B rays with externally administered ABA are shown in a boxed line graph, * indicates *p* < 0.05.

**Figure 4 biomolecules-13-01700-f004:**
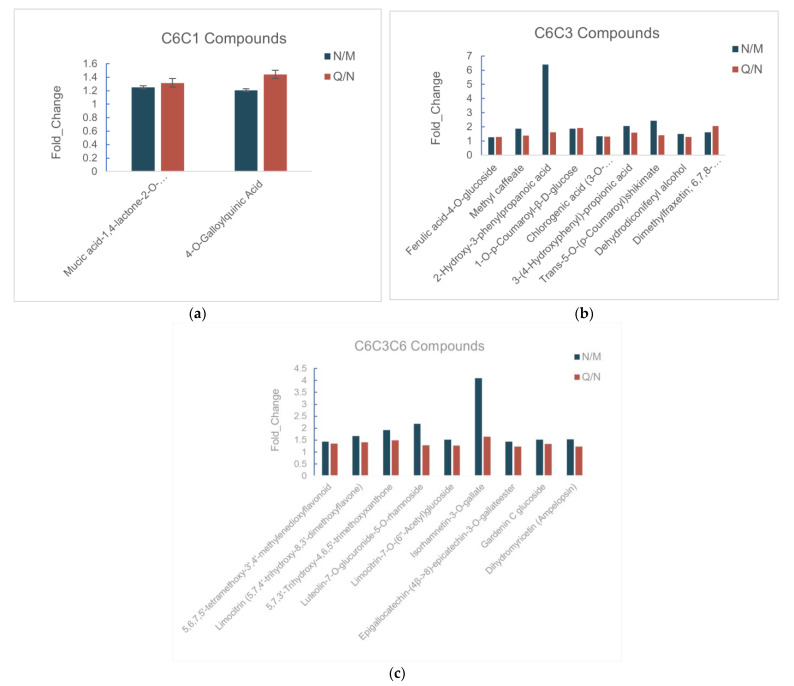
Statistics of phenolic and flavonoid abundance under the effects of UV-B and UV-B treatment with externally administered ABA. The proportion of each substance’s expression is set as the vertical coordinate. (**a**) C6C1-carbon skeleton phenols; (**b**) phenolics with the carbon skeleton C6C3; (**c**) phenolics with the carbon skeleton C6C3C6.

**Figure 5 biomolecules-13-01700-f005:**
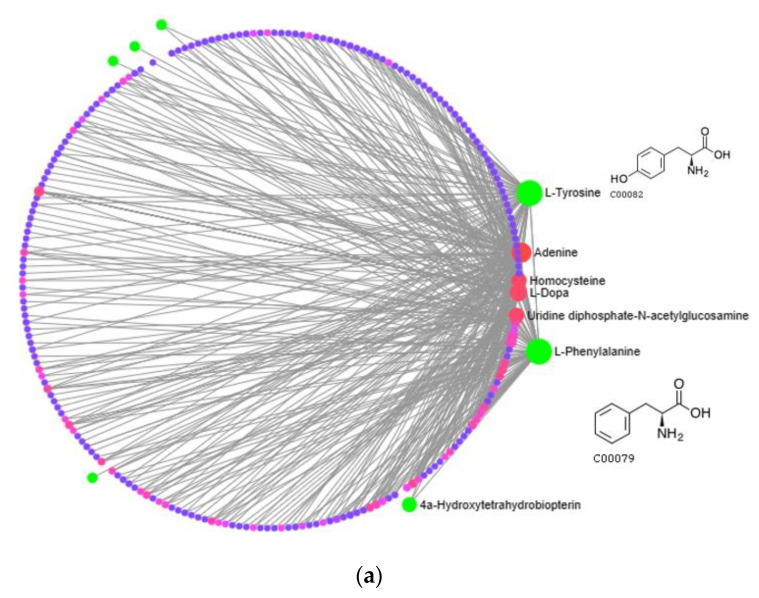
Network analysis and correlation of different primary and secondary metabolites linked to *R. chrysanthum*’s reaction to UV-B stress are shown. (**a**) Degrees are the number of connections to other points, and green is used to highlight the substances that are the focus of the study. (**b**–**d**) Correlation analysis of amino acids and their derivatives with phenolic acids, flavonoids, and lignans, alkaloids, and terpenoids, respectively. (**e**–**g**) Correlation analysis of nucleotides and their derivatives with phenolic acids, flavonoids, and lignans, alkaloids, terpenoids, respectively. (**h**–**j**) Correlation analysis of sugars, vitamins, and free fatty acids with phenolic acids, flavonoids, and lignans, alkaloids, and terpenoids, respectively.

**Figure 6 biomolecules-13-01700-f006:**
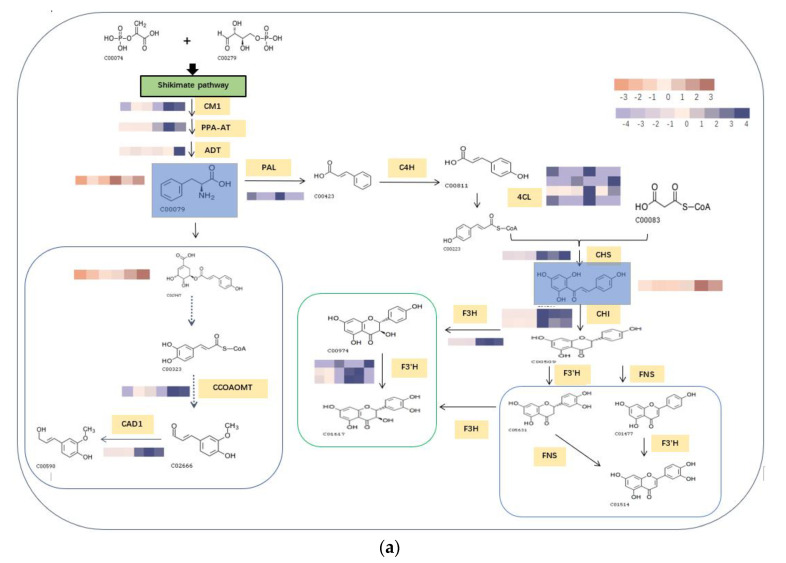
Visualization of dynamic pathways of primary and phenolic metabolites. (**a**) Yellow fills indicate enzymes that cause changes in the pathway, while green fills indicate metabolites with significant changes in the pathway. The flavanone metabolic pathway is represented by pink boxes, the flavonol metabolic pathway by green boxes, and the monolignol metabolic pathway by blue boxes. In the heat map, the color pink denotes that UV-B duress has almost no impact on the enzyme, the color light purple denotes that UV-B radiation inhibits the enzyme’s activity, and the color dark purple denotes that UV-B radiation increases the enzyme’s activity; likewise, a metabolite’s light brown hue suggests that UV-B radiation is downregulating it, whereas a metabolite’s dark brown color indicates that UV-B radiation is upregulating it. (**b**–**d**) L-phenylalanine, naringenin chalcone, and Trans-5-O-(p-Coumaroyl) shikimate all show significant alterations. Significant difference (* *p* < 0.05).

**Figure 7 biomolecules-13-01700-f007:**
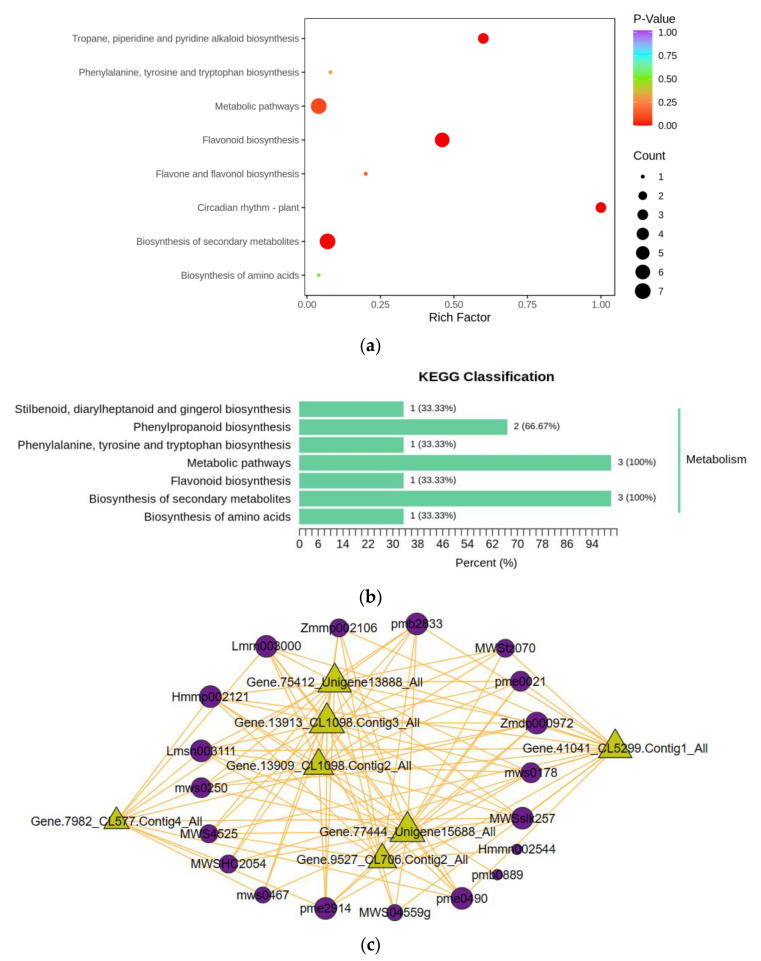
Key enzymes involved in *R. chrysanthum*’s reaction to UV-B exposure were analyzed using the KEGG pathway method, and correlations between key enzymes and metabolites were examined. (**a**,**b**) Key enzymes’ KEGG bubble and bar graphs; (**c**) links between proteins and metabolomes. Proteins are represented by triangles, metabolites by circles, and positive relationships by solid yellow lines.

**Table 1 biomolecules-13-01700-t001:** The first seven key core DMs’ VIP and *p*-value changes in different components under UV-B circumstances of stress.

Index	Structural Formula	Compounds	VIP (MN)	*p*-Value (MN)	VIP (NQ)	*p*-Value (NQ)
pme0490	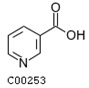	Nicotinic acid (Vitamin B3)	1.707	0.021	1.290	0.182
Zmdp000972	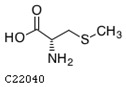	S-methyl-L-cysteine	1.683	0.010	1.610	0.078
pme2914	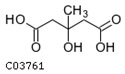	3-hydroxy-3-methylpentane-1,5-dioic acid	1.676	0.012	1.425	0.097
mws0250	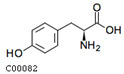	L-tyrosine*	1.655	0.013	1.627	0.079
pme0021	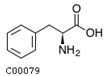	L-phenylalanine	1.655	0.017	1.694	0.057
MWS4525	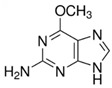	6-O-methylguanine	1.602	0.029	1.268	0.166
MWSslk257	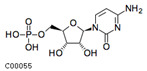	Cytidine 5′-monophosphate(Cytidylic acid)	1.595	0.112	1.268	0.192

**Table 2 biomolecules-13-01700-t002:** Information about 11 important enzymes under UV-B stress.

Protein Accession	KEGG KO No.	KEGG Gene	M/N Ratio
Gene.41041_CL5299.Contig1_All	K00475	F3H; naringenin 3-dioxygenase [EC:1.14.11.9]	0.573
Gene.13909_CL1098.Contig2_All	K00660	CHS; chalcone synthase [EC:2.3.1.74]	0.48
Gene.75412_Unigene13888_All	K00660	CHS; chalcone synthase [EC:2.3.1.74]	0.546
Gene.13913_CL1098.Contig3_All	K00660	CHS; chalcone synthase [EC:2.3.1.74]	0.557
Gene.22273_CL2057.Contig1_All	K01850	E5.4.99.5; chorismate mutase [EC:5.4.99.5]	0.727
Gene.77444_Unigene15688_All	K01859	E5.5.1.6; chalcone isomerase [EC:5.5.1.6]	0.781
Gene.9527_CL706.Contig2_All	K01859	E5.5.1.6; chalcone isomerase [EC:5.5.1.6]	0.865

## Data Availability

The datasets generated during and/or analyzed during the current study are available from the corresponding author on reasonable request.

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
