# Peer review of "Rhododendron chrysanthum’s Primary Metabolites Are Converted to Phenolics More Quickly When Exposed to UV-B Radiation"

_biomolecules, 2023, doi:10.3390/biom13121700_

Round 1

Reviewer 1 Report

Comments and Suggestions for Authors

The manuscript addresses a topic of interest to the journal's readers. It presents interesting results in the area, from my point of view it is only necessary to review some aspects of the manuscript, such as: That the scientific names be duly reported and in italics. Avoid double spaces as well as double characters (parentheses, signs, etc.) The results provided by the authors are coarse, however, the conclusions are too general. It is advisable to expand these based on the results shown or to specify a little more based on the results. Figure 7., sections a and b, it is recommended to enlarge the image for greater clarity. The same comment for Figure 3, Section d.

Reviewer 2 Report

Comments and Suggestions for Authors

The manuscript “Rhododendron chrysanthum's primary metabolites are converted to phenolics more quickly when exposed to UV-B radiation” is interesting and highlights the UV-B irradiation effect on plant metabolites. Positive and negative correlations between amino acids and their derivatives with phenolic compounds are established. . I have minor requests for revision:

·          After the title, omit the punctuation.

·         Please delete the template text in lines 475–479.

Author Response

Dear reviewer:
Thank you very much for your careful review of the content of my manuscript. I am honored to receive your approval, and at the same time, your suggestions have helped me to identify shortcomings in the preparation of my manuscript. I thank you for your suggestions on the format of my manuscript, and I have now made changes to it according to your requirements:

  • Punctuation after the title has been removed
  • The template text in lines 475-479 has been deleted.

Reviewer 3 Report

Comments and Suggestions for Authors

This paper reports on the «Rhododendron chrysanthum's primary metabolites are converted to phenolics more quickly when exposed to UV-B radiation». This study is original and interesting. However, certain aspects of the study require further refinement. The following observations and suggestions are offered for improvement:

Abstract:

Enhance the abstract by incorporating application-type conclusions at the end to provide a more impactful summary.

Verify the consistent use of "UV-B" throughout the manuscript at lines 22, 253, 258, 285, 392, 468, and 473.

Keywords:

phenolic compounds

Consider adding ABA to the list of keywords for a more comprehensive representation of the study.

Introduction:

Line 60: The mention of phenolics in this sentence is redundant.

Integrate ABA into the introduction, providing relevant background information and elucidating its role within the hypothesis.

Eliminate redundancy in lines 53 and 57 regarding the quantity of data: “more than 8,000”

Materiales y Métodos:

Do you have an ABA control group?

Ensure uniformity in the use of units (hours, minutes, liters, meters).

Line 134: Check units 1200 l

Line 137: Check units 0.22 m

Address the repetition of information in lines 135-137 and verify the accuracy of the mention of "Rhododendron bovis" in line 138.

Did you perform the entire proteome analysis methodology with the same conditions as reference 30?

Figures:

Improve the quality of Figures 1, 3, 6, and 7 for better visibility. Add a brief explanation of the abbreviation DMs in the main text for clarity.

Discussion:

Delete lines 476-478 from the discussion section.

Conclusions:

Include observations regarding ABA in the conclusions. Provide insights into future research possibilities and the practical applications of the obtained results, emphasizing their importance.

Addressing the outlined points will contribute to the overall clarity and scientific rigor of the paper.
